# Effects of Dietary Zeolite Supplementation as an Antibiotic Alternative on Growth Performance, Intestinal Integrity, and Cecal Antibiotic Resistance Genes Abundances of Broilers

**DOI:** 10.3390/ani9110909

**Published:** 2019-11-01

**Authors:** Hengman Qu, Yefei Cheng, Yueping Chen, Jun Li, Yurui Zhao, Yanmin Zhou

**Affiliations:** College of Animal Science and Technology, Nanjing Agricultural University, No. 6, Tongwei Road, Xuanwu District, Nanjing 210095, China; qhm13457@163.com (H.Q.); 2017205020@njau.edu.cn (Y.C.); chenyp0321@163.com (Y.C.); 2017105070@njau.edu.cn (J.L.);

**Keywords:** zeolite, antibiotic, growth performance, intestinal barrier, antibiotic resistance genes, broilers

## Abstract

**Simple Summary:**

The widespread use of antibiotics in livestock production has attracted public attention due to antibiotic resistance. It is necessary to search for antibiotic alternatives in feeds. Zeolite, a kind of feed additive or feed raw material, can enhance growth performance, balance intestinal bacteria, and improve antioxidant capacity in animals. *In vitro* studies showed zeolite could reduce antibiotic resistance genes (ARGs) abundances. This research compared the effects of antibiotics and zeolite on growth performance, intestinal morphology and barrier function, and cecal ARGs abundances of broilers. Our results indicated that dietary zeolite supplementation exerted similar or better effects than antibiotic inclusion.

**Abstract:**

The study investigated the effects of dietary zeolite supplementation as an antibiotic alternative on growth performance, intestinal integrity, and cecal antibiotic resistance genes abundances of broilers. One-day-old chicks were assigned into three groups and fed a basal diet or a basal diet supplemented with antibiotics (50 mg/kg) or zeolite (10 g/kg). Antibiotic or zeolite increased (*p* < 0.05) average daily gain (ADG) from 1 to 42 days and duodenal villus height to crypt depth ratio (VH:CD) at 21 days. Zeolite increased (*p* < 0.05) ADG and average daily feed intake from 1 to 21 days, jejunal VH:CD at 21 and 42 days, ileal VH and VH:CD at 42 days, zonula occludens-1 mRNA abundance at 21 days, and duodenal occludin mRNA abundance at 42 days, whereas reduced (*p* < 0.05) jejunal CD and malondialdehyde levels in ileum at 21 days and duodenum at 42 days, serum D-lactic acid and diamine oxidase levels at 42 days, and plasma lipopolysaccharide content at 21 and 42 days. Antibiotics reduced (*p* < 0.05) duodenal claudin-2 mRNA abundance at 21 days, whereas increased (*p* < 0.05) cecal *tetB* abundance at 42 days. These findings suggested that the beneficial effects of zeolite in broilers were more pronounced than that of antibiotics.

## 1. Introduction

The widespread use of in-feed antibiotics has arisen into a controversial issue worldwide and is facing reduced social acceptance due to the public health problem, such as antibiotic residue and antibiotic resistant bacteria [1,2,3,4]. The production of antibiotic resistance genes (ARGs) is an important reason for bacterial resistance and has become an emerging environmental contaminant [5]. It was investigated that more than 95% *Escherichia coli* isolates from sick chickens were resistant to at least one antibiotic between 1993 and 2013 in China [6]. Thus, using antibiotic growth promoters has been eliminated or limited. On 1 January 2006, the European Union withdrew the approval for antibiotics as growth promoters [7]. Recently, the Ministry of Agriculture of China will prohibit the use of all kinds of antibiotics as growth promoters from 1 January 2020 [8]. As a result, it is necessary to find potential alternatives to in-feed antibiotics in animals’ production. 

Zeolite is crystalline, hydrated aluminosilicates of alkali and alkaline earth cations, which has a large number of regularly arranged, uniform size, interlinked pores and channels, large specific surface area, strong adsorption, ion exchangeability, and good heat and acid resistance [9,10,11]. These characteristics of zeolite endow its extensive applications in various industries, including metallurgy [12], pesticide [13], environmental protection [14,15], and livestock production [16,17]. As a kind of feed additive or feed raw material, several researches have shown that zeolite promoted growth performance [10,18,19,20], improved digestive function [20,21], absorbed harmful substances in digestive tract [20,22], benefited intestinal microecology and villus development [23,24], and enhanced antioxidant capacity [16,17] in broilers and/or pigs. Interestingly, it is reported that zeolite exerted a similar effect of antibiotics, efficiently working against the gram-negative bacteria [25,26,27,28]. Moreover, *in vitro* studies have also shown that zeolite can absorb antibiotics strongly [29,30] and even reduce the abundances of some ARGs in sludge compost [31]. Poultry production is currently perceived as one of the major sources of pathogens and antimicrobial resistance [32]. However, whether zeolite can reduce *in vivo* ARGs produced during growth and development in broilers is still unknown. Based on the aforementioned, we speculated that dietary zeolite supplementation might be an additive comparable to antibiotics in feed. The current study was, therefore, designed to evaluate the effects of zeolite and antibiotics on growth performance, intestinal barrier function and integrity, and ARGs abundance in the cecal contents of broilers, and hence to investigate whether dietary zeolite supplementation can be an additive that could produce more pronounced beneficial consequences than antibiotics used in broilers diets.

## 2. Materials and Methods 

### 2.1. Animals, Diets, and Treatments

All experimental conditions and animal procedures were approved by the Nanjing Agricultural University Institutional Animal Care and Use Committee (Certification No. SYXK (Su) 2017-0007).

A total of 144, 1-day-old male Arbor Acres broilers with similar hatching weight were obtained from a commercial hatchery and raised from 1 to 42 days. The chicks were randomly distributed into 3 dietary treatments consisting of 6 replicates. A replicate included a cage with 8 chicks. The chicks in 3 treatments were received a corn-soybean meal basal diet (CON), and the basal diet supplemented with either 50 mg/kg chlortetracycline (ANT, Jinhe Biotechnology Co. Ltd. Hohhot, China) or 10 g/kg zeolite (ZEO), which was kindly gifted by the Beipiao Xinmingshuang Zeolite Agency (Beipiao, China). Using a dosage at 10 g/kg was according to these studies that showed that 10 g/kg zeolite supplementation exhibited beneficial consequences in improving the growth performance and intestinal flora composition of broilers [33,34,35]. Ingredient compositions and nutrient contents of the basal diet are the same as the study of Cheng et al. [36]. Birds were allowed free access to water and mash feed in 3-level cages (120 cm × 60 cm × 50 cm) in a room with controlled environmental conditions. Continuous light in the room was provided during the whole experimental period. The temperature of the room was maintained at 32 to 34 °C for the first 3 days and then reduced by 2 to 3 °C per week to a final temperature of 20 °C. At 21 and 42 days of age, birds were weighed after a 12-h feed withdrawal, and feed intake was recorded by replicate (cage) to calculate average daily gain (ADG), average daily feed intake (ADFI), and the feed/gain ratio (F/G). Birds that died during the experimental period were weighed, and the data were included in the calculation of F/G.

### 2.2. Sample Collection

At days 21 and 42 of the experiment, 18 male birds (one bird close the average body weight of each replicate) were selected. Whole blood samples were collected in both glass anti-coagulant tubes coated with heparin sodium (Xiamen Bioendo Technology Co., Ltd. Xiamen, China) and tubes without anti-coagulant by wing vein puncture. Plasma separated from blood samples collected in anti-coagulant tubes after centrifugation at 3000× g for 3 min at 4 °C. Serum was separated from blood samples collected in tubes without anti-coagulant after centrifugation at 4450× g for 15 min at 4 °C. The plasma and serum were immediately frozen at −20 °C until subsequent analysis. Then, birds were euthanatized by cervical dislocation and necropsied immediately. The whole gastrointestinal tract was rapidly removed and separated. About 2 cm segments of mid-duodenum, mid-jejunum, and mid-ileum were then harvested, flushed repeatedly with ice-cold PBS (pH 7.4), and immediately immersed in 10% fresh, chilled paraformaldehyde solution for subsequent histological measurement. Then, the mucosa of duodenum, jejunum, and ileum was collected and rapidly frozen in liquid nitrogen and stored at −80 °C for further analysis. The cecum samples (left side) were excised aseptically, and the contents were removed to a sterile cryogenic vial rapidly and frozen in liquid nitrogen and stored at −80 °C for further analysis.

### 2.3. Morphological Examination

Intestinal segments were dehydrated, cleared, and embedded in paraffin. Serial sections were then cut at 5 µm, deparaffinized in xylene, rehydrated, and stained with hematoxylin and eosin. Histological slides were prepared from 3 cross-sections of each intestinal sample. A total of 10 intact, well-oriented crypt-villus and associated crypts were measured from each segment. Villus height and crypt depth were measured using a Nikon ECLIPSE 80i light microscope equipped with a computer-assisted morphometric system (Nikon Corporation, Tokyo, Japan). All staining chemicals were sourced from Sigma-Aldrich Chemical (St Louis, MO, USA).

### 2.4. Analysis of Mucosal Antioxidant Parameters

Approximately 0.3 g of duodenal, jejunal, and ileal mucosa samples were homogenized (1:9, wt/vol) with ice-cold 154 mmol/L sodium chloride solution using an Ultra-Turrax homogenizer (Tekmar Co, Cincinnati, OH, USA) and then centrifuged at 4450× g for 15 min at 4 °C. The supernatant was then collected and stored at −20 °C for assaying mucosal antioxidant and immune parameters. Total protein concentration, malondialdehyde (MDA) content, total antioxidant capacity (T-AOC), and total superoxide dismutase (T-SOD) activity were measured using diagnostic kits (Nanjing Jiancheng Bioengineering Institute, Nanjing, China) according to the manufacturer’s instructions. All results were normalized against total protein concentration in each sample for inter-sample comparison.

### 2.5. Measurement of Bacterial Lipopolysaccharide (LPS) Content in Plasma and D-Lactic Acid Level and Diamine Oxidase Activity (DAO) in Serum

The plasma LPS content was determined by a chromogenic matrix Limulus kit (Xiamen Bioendo Technology Co., Ltd. Xiamen, China) according to the manufacture’s instruction. The serum D-lactic acid concentration was quantified using a D-lactic acid colorimetric assay kit (BioVision Inc. Shanghai, China) on the basis of the manufacturer’s protocols. The activity of DAO in the serum was determined by a corresponding reagent kit (Nanjing Jiancheng Bioengineering Institute, Nanjing, China). 

### 2.6. Messenger RNA Quantitative PCR

Total RNA from intestinal mucosa was isolated using Trizol reagent according to the instructions of the manufacturer (TaKaRa Biotechnology Co. Ltd., Dalian, China). The final concentration and purity of RNA was quantified using a NanoDrop ND-1000 UV spectrophotometer (Nano Drop Technologies, Wilmington, DE, USA) from OD260/280 readings obtaining ratios between 1.8 and 2.1. RNA samples were then diluted with diethyl pyrocarbonate-treated water (Biosharp) to a final concentration of 0.5 µg/µL. After that, 1 µg of total RNA was reversed immediately following the RNA isolation using the Prime Script TM RT reagent kit (TaKaRa Biotechnology Co., Ltd., Dalian, China). The primer sequences were the same with our present study [37] and synthesized by Invitrogen Biotechnology Co., Ltd. (Shanghai, China). The cDNA samples were amplified with the TB Green™ Premix Ex Taq™ kit (Takara Biotechnology Co. Ltd., Dalian, China) based on an ABI StepOne Plus real-time PCR system (Applied Biosystems, Foster City, CA, USA). Detailed procedures of real-time quantitative PCR were performed following the descriptions by our present study [37]. The gene expression level was calculated relative to β-actin using the 2^−∆∆CT^ method [38].

### 2.7. DNA Extraction and Quantitative PCR (qPCR)

DNA was extracted from 0.2 g of each cecal contents sample using the QIAamp Fast DNA Stool Mini Kit for feces (QIAGEN GmbH, Hilden, Germany) according to the manufacturer’s instructions. The 16S rRNA gene and 11 tetracycline resistance genes (*tetA*, *tetB*, *tetC*, *tetE*, *tetG*, *tetM*, *tetO*, *tetQ*, *tetT*, *tetW*, and *tetX*) were detected and analyzed in this study. All qPCR assays were performed using the TB Green™ Premix Ex Taq™ kit (Takara Biotechnology Co. Ltd., Dalian, China) on the ABI StepOne Plus real-time PCR system (Applied Biosystems, Foster City, CA, USA). The qPCR mixture of the 11 genes were in a total volume of 10 μL, containing 5 μL TB Green Premix Ex Taq II (TliRNaseH Plus) (2×) (Takara Biotech, Co., Ltd., Dalian, China), 0.4 μL of each primer (primer sequences are referenced the study of Zhu et al. [39]), 0.2 μL ROX Reference Dye (50×), and 1 μL of DNA template, and the rest of the volume was made up of distillation-distillation H_2_O. The thermal cycling steps for quantitative PCR amplification were as follows: (1) 95 °C for 30 s; (2) 95 °C for 5 s; (3) annealing temperature at 60 °C for 30 s, where steps (2) to (3) were repeated 40 times. The relative abundances of ARGs were calculated relative to 16S rRNA using the 2^−∆∆CT^ method [38].

### 2.8. Statistical Analysis

Data were analyzed by one-way analysis of variance (ANOVA) using the SPSS statistical software (Ver.16.0 for windows, SPSS Inc., Chicago, IL, USA) with a pen (cage) as the experimental unit. Differences among treatments were examined using the Tukey-Kramer’s multiple range tests, which were considered significant when the *p*-value was less than 0.05. The means and standard errors of means (SEM) were presented.

## 3. Results

### 3.1. Growth Performance

Compared with the CON group (Table 1), dietary zeolite supplementation increased the ADG and ADFI of broilers from 1 to 21 days (*p* < 0.05). Moreover, dietary either zeolite or antibiotic supplementation improved ADG of broilers from 1 to 42 days (*p* < 0.05), with the values of ADG being similar between the two treatments (*p* > 0.05). However, treatments did not affect growth performance from 22 to 42 days (*p* > 0.05).

### 3.2. Intestinal Morphology

Compared with the CON group (Table 2), birds fed either zeolite or antibiotic had a higher duodenal VH:CD at 21 days (*p* < 0.05). Dietary zeolite supplementation increased (*p* < 0.05) ileal VH at 42 days, VH:CD in the jejunum at both 21 and 42 days, and ileum at 42 days, whereas decreased (*p* < 0.05) jejunal CD at 42 days. Additionally, dietary antibiotic supplementation reduced jejunal CD at 21 days when in comparison with the CON group (*p* < 0.05); however, zeolite inclusion did not exert such effect (*p* > 0.05).

### 3.3. Antioxidant Function of Intestinal Mucosa

As shown in Table 3, compared with the CON group, dietary zeolite supplementation reduced (*p* < 0.05) MDA content in the ileum at 21 days and duodenum at 42 days. However, this effect was not observed by antibiotic inclusion (*p* > 0.05). Antioxidant enzyme activities were similar among groups (*p* > 0.05). 

### 3.4. Plasma LPS Content and Serum DAO and D-Lactic Levels 

Compared with the CON group (Table 4), dietary zeolite supplementation reduced (*p* < 0.05) plasma LPS content at 21 days and serum DAO and D-lactic acid level at 42 days, whereas the values of these indexes were not affected by antibiotic supplementation (*p* > 0.05). Moreover, dietary zeolite supplementation decreased (*p* < 0.05) plasma LPS concentration at 42 days compared with the ANT group.

### 3.5. Gene Expression Levels Related to Intestinal Barrier Function

Broilers receiving zeolite exhibited higher mRNA abundances of zonula occludens-1 (*ZO-1*) in the jejunum at 21 days and occludin (*OCLN*) in the duodenum at 42 days compared with those fed the basal diet (*p* < 0.05). In contrast, dietary antibiotic supplementation reduced claudin-2 (*CLDN2*) gene expression level in the duodenum at 21 days (*p* < 0.05). Treatments did not affect intestinal mucosal claudin-3 (*CLDN3*) mRNA abundance (*p* > 0.05, Table 5).

### 3.6. ARGs Detection Frequencies (DFs) in the Cecal Content

Compared with the CON group (Table 6), AGRs detection frequencies of cecal contents in the ANT group were the highest. The *tetG*, *tetM*, *tetO*, *tetQ,* and *tetW* were detected in all samples at 21 days. *TetB*, *tetC*, *tetM*, *tetO*, *tetQ*, *tetW,* and *tetX* were detected in all samples at 42 days. However, detection frequencies of *tetA*, *tetE,* and *tetT* were too low; thus, they were not assayed in the further. 

### 3.7. Relative Abundances of ARGs in the Cecal Content

Compared with the CON group, dietary antibiotic supplementation observably increased the relative abundance of *tetB* in cecal contents of broilers at 42 days (*p* < 0.05, Table 7). However, dietary zeolite supplementation did not upregulate ARGs relative abundances in the cecal contents (*p* > 0.05). In contrast, broilers fed zeolite exerted lower relative abundances of *tetB*, *tetC,* and *tetX* in the cecal contents at 42 days when compared with the ANT group (*p* < 0.05). 

## 4. Discussion

In this study, broilers fed the basal diet supplemented with zeolite significantly exhibited an increase in ADG and ADFI from days 1 to 21 and ADG from 1 to 42 days, which had better effects than antibiotic supplementation. The improved growth performance of broilers was also demonstrated by Wu et al. [16,17]. Additionally, the dietary inclusion of zeolites can improve ADG and/or feed conversion ratio in pigs [10,20], calves [10], and sheep [10]. The beneficial consequences of zeolite on animal growth performance could be attributed to its binding effect of ammonia [40], absorption of toxins produced during intestinal microbial degradation [41], retarding effect on the digesta transit [10], regulation on digestive enzyme activity [42], and adsorption of mycotoxins [43]. 

The normal microarchitecture of the small intestine is very crucial in maintaining nutrient absorption and resistance to harmful substrates [44,45,46,47]. The VH:CD is considered to be an important criterion for estimating the digestive capacity and absorptive function of the small intestine [16,17]. In the current study, zeolite supplementation exerted a positive influence on intestinal morphology, being embodied in increased VH:CD in the duodenum and jejunum at 21 days, in the jejunum and ileum at 42 days, and VH in the ileum at 42 days, as well as decreased CD in the jejunum at 42 days. Likewise, it has been previously reported that dietary zeolite supplementation could improve intestinal integrity through increasing VH in the jejunum and ileum of female broilers [45]. In addition, palygorskite, with similar characteristics to zeolite, was demonstrated by our lab team members that it could enhance intestinal morphology in broilers and laying hens [47,48,49]. This may because zeolite can adsorb intestinal pathogens bacterium, prevent intestinal mucosa from damage, reinforce the intestinal mucosal barrier, and help in the regeneration of the epithelium [26,50]. Furthermore, the ion-exchange properties of minerals can alter the enzymatic activity of intestinal secretions, which could stimulate the villi and microvilli of intestinal mucosa [51]. Therefore, dietary zeolite supplementation would be an efficient way to improve intestinal development in broilers. 

The reactive oxygen species (ROS) are produced during normal metabolism in cells [52]. However, the overproduction of ROS during cellular metabolism could result in oxidative damage [53]. The MDA is the main end-product of the lipid peroxidation caused by ROS, and the accumulation of MDA is usually considered a marker of lipid peroxidation [54]. It is reported that zeolite can decrease the MDA content in the jejunum and ileum of broilers [16,17]. Consistently, in the present work, broilers receiving zeolite exhibited decreased MDA content in the ileum at 21 days and in the duodenum at 42 days. The improvement of zeolite on antioxidant capacity in broilers may be partly attributed to its adsorption effect, resulting in reduced oxidative stress by pathogens and other hazardous substances in the gut. In addition, the enhanced intestinal integrity of broilers given zeolite in this study would contribute to a better intestinal health status, which is also a nonnegligible factor for improved antioxidant capacity in broilers. 

The DAO is an intracellular enzyme that exists in intestinal epithelial villus cells [55]. Bacterial lipopolysaccharide (LPS) is the major component of the outer surface membrane in almost all gram-negative bacteria [56]. D-lactic acid is a product of bacterial fermentation of carbohydrates, which can be considered as a sensitive marker for reflecting intestinal injury and monitoring intestinal permeability [57]. Decreased DAO activity and contents of D-lactic acid and LPS are usually associated with improved intestinal barrier function of animals [58,59]. In the present study, zeolite supplementation significantly decreased LPS concentrations in plasma at 21 days and DAO activity and D-lactic acid level in serum at 42 days. Tight junctions are the crucial components of the intestinal mucosal barrier. They mainly consist of peripheral membrane protein ZO-1 and the transmembrane protein OCLN and claudins [60]. Herein, the upregulation of *ZO-1*, *OCLN*, and claudins mRNA expression levels would be beneficial to intestinal structure and barrier function. In this work, increased *ZO-1* gene expression level in jejunal mucosa at 21 days and *OCLN* mRNA abundance in duodenal mucosa at 42 days were observed by zeolite supplementation. Those coupled with the simultaneously decreased circulating DAO activity, D-lactic acid, and LPS level, further indicated that zeolite could improve the intestinal barrier function in broilers. Moreover, palygorskite clay with similar characteristics was illustrated to enhance intestinal barrier function in broilers and laying hens, as evidenced by decreased serum DAO activity and gene expression levels related to intestinal barrier function [43,44,45,61]. The improved intestinal morphology and barrier function resulting from zeolite supplementation in the present study indicated that dietary zeolite supplementation could improve intestinal integrity in broilers. On the one hand, zeolite has a strong adsorptive capacity and can adsorb toxic substances and bacteria in the gut that is harmful to the host, eventually excreting them from animals’ bodies [16,17,62,63]. On the other hand, zeolite has a big specific surface area and allows it to be distributed evenly on the surface of the intestinal mucosa, therefore forming a protective screen to attenuate the gut damage by harmful substances and bacteria [62,64].

The presence of ARGs is the primary reason for bacteria becoming resistant to antibiotics and has become one of the most important public health concerns around the world [5]. Animal intestinal microorganisms live under the pressure of antibiotics when fed a diet supplemented with antibiotics for a long time, and they are prone to develop antibiotic resistance and generate ARGs [1,2,3,4,5]. The results showed that the detection frequencies of *tetA*, *tetE,* and *tetT* were less than 100%, whereas frequencies of the other eight genes were detected 100% in the cecal content (21 or 42 days), so, we then selected the eight *tet* genes (*tetB*, *tetC*, *tetG*, *tetM*, *tetO*, *tetQ*, *tetW,* and *tetX*) for further study. As expected, our experimental results showed that antibiotic inclusion increased the detection rate and relative abundances of ARGs (*tetB*, *tetC,* and *tetX*) in the cecal contents of broilers. It is previously demonstrated that zeolite can reduce the abundances of some ARGs in sludge compost *in vitro* [31]. In this work, dietary supplementation of zeolite significantly decreased cecal *tetB*, *tetC,* and *tetX* relative abundances compared with the ANT group, and also numerically reduced their abundances when compared with the CON group. This may be due to its sporous structure and the ability to reduce the selective pressure from heavy metals and the rate of microbe contact, which then causes the horizontal gene transfer through conjugation to reduce [31]. These results indicated that dietary zeolite inclusion could decrease ARGs relative abundances in the cecal content of broilers to some extent, which may be because zeolite can adhere or kill harmful bacteria and herein reducing a possibility of gut bacteria producing ARGs.

## 5. Conclusions

The study suggested that dietary zeolite supplementation had better effects on growth performance, intestinal oxidative status, and integrity in broilers than antibiotic supplementation. Furthermore, zeolite did not increase cecal ARGs abundance. Therefore, the beneficial effects of dietary zeolite supplementation in broilers are more pronounced than the antibiotic used in feed in this study.

## Figures and Tables

**Table 1 animals-09-00909-t001:** Effects of zeolite and antibiotic supplementation on growth performance in broilers.

Items ^1,2^	CON	ANT	ZEO	SEM	*p*-Value
1–21 days					
ADG (g/d)	26.04 ^b^	28.01 ^ab^	30.26 ^a^	0.716	0.009
ADFI (g/d)	39.98 ^b^	41.84 ^ab^	45.07 ^a^	0.743	0.008
F/G (g/g)	1.54	1.50	1.49	0.016	0.495
22–42 days					
ADG (g/d)	73.43	77.17	75.64	1.131	0.423
ADFI (g/d)	155.13	159.09	151.54	2.348	0.428
F/G (g/g)	2.05	2.07	2.01	0.028	0.657
1–42 days					
ADG (g/d)	49.73 ^b^	52.58 ^a^	52.80 ^a^	0.529	0.019
ADFI (g/d)	93.68	97.91	96.95	1.153	0.308
F/G (g/g)	1.88	1.86	1.84	0.017	0.591

^a, b^ Means within a row with different superscripts differ significantly at *p* < 0.05. ^1^ ADG, average daily gain; ADFI, average daily feed intake; F/G, feed to gain ratio. ^2^ CON: broilers receiving a corn-soybean meal basal diet without antibiotic and zeolite; ANT: broilers receiving the basal diet supplemented with 50 mg/kg chlortetracycline; ZEO: broilers were given the basal diet supplemented with 10 g/kg zeolite. SEM, standard error of means (n = 6).

**Table 2 animals-09-00909-t002:** Effects of zeolite and antibiotic supplementation on intestinal integrity in broilers.

Items ^1,2^	CON	ANT	ZEO	SEM	*p*-Value
21 days					
Duodenum					
VH (µm)	1661.18	1728.58	1663.85	42.077	0.782
CD (µm)	232.44	190.98	184.59	9.255	0.063
VH:CD	7.26 ^b^	9.11 ^a^	9.25 ^a^	0.347	0.021
Jejunum					
VH (µm)	1282.88	1182.81	1441.12	52.896	0.131
CD (µm)	274.70 ^a^	212.79 ^b^	231.84 ^ab^	10.820	0.046
VH:CD	4.73 ^b^	5.55 ^ab^	6.38 ^a^	0.262	0.026
Ileum					
VH (µm)	825.50	789.28	748.92	40.177	0.762
CD (µm)	159.77	147.93	152.37	8.823	0.872
VH:CD	5.32	5.82	4.91	0.301	0.491
42 days					
Duodenum					
VH (µm)	1922.33	1752.94	1841.45	33.717	0.118
CD (µm)	239.03	207.98	207.25	7.190	0.116
VH:CD	8.16	8.54	8.89	0.215	0.406
Jejunum					
VH (µm)	1353.73	1464.09	1410.14	48.407	0.675
CD (µm)	194.61 ^a^	173.01 ^ab^	131.97 ^b^	10.082	0.024
VH:CD	7.18 ^b^	8.51 ^b^	10.69 ^a^	0.410	<0.001
Ileum					
VH (µm)	824.86 ^b^	863.51 ^ab^	1057.39 ^a^	38.246	0.018
CD (µm)	131.93	117.62	115.79	3.852	0.177
VH:CD	6.25 ^b^	7.52 ^ab^	9.19 ^a^	0.405	0.004

^a, b^ Means within a row with different superscripts differ significantly at *p* < 0.05. ^1^ VH, villus height; CD, crypt depth; VH:CD, the ratio of villus height to crypt depth. ^2^ CON: broilers receiving a corn-soybean meal basal diet without antibiotic and zeolite; ANT: broilers receiving the basal diet supplemented with 50 mg/kg chlortetracycline; ZEO: broilers were given the basal diet supplemented with 10 g/kg zeolite. SEM, standard error of means (n = 6).

**Table 3 animals-09-00909-t003:** Effects of zeolite and antibiotic supplementation on antioxidant function of intestinal mucosa in broilers.

Items ^1,2^	CON	ANT	ZEO	SEM	*p*-Value
21 days					
Duodenum					
MDA (nmol/mg protein)	0.73	0.62	0.64	0.059	0.740
T-AOC (U/ mg protein)	0.55	0.46	0.63	0.032	0.103
T-SOD (U/mg protein)	190.17	190.28	224.43	7.684	0.125
Jejunum					
MDA (nmol/mg protein)	1.03	0.77	0.71	0.059	0.054
T-AOC (U/mg protein)	0.56	0.61	0.65	0.020	0.212
T-SOD (U/mg protein)	190.61	190.08	199.62	2.717	0.389
Ileum					
MDA (nmol/mg protein)	1.30 ^a^	1.34 ^a^	0.73 ^b^	0.093	0.006
T-AOC (U/mg protein)	1.01	1.03	1.16	0.058	0.541
SOD (U/mg protein)	172.50	170.73	174.51	5.528	0.968
42 days					
Duodenum					
MDA (nmol/mg protein)	1.32 ^a^	1.08 ^a^	0.76 ^b^	0.071	0.001
T-AOC (U/ mg protein)	0.87	0.89	0.90	0.035	0.944
T-SOD (U/mg protein)	211.36	215.05	213.97	6.410	0.974
Jejunum					
MDA (nmol/mg protein)	0.71	0.64	0.62	0.044	0.733
T-AOC (U/ mg protein)	0.78	0.78	0.83	0.030	0.791
T-SOD (U/mg protein)	225.15	238.01	229.00	7.015	0.758
Ileum					
MDA (nmol/mg protein)	0.95	0.72	0.80	0.049	0.240
T-AOC (U/mg protein)	1.09	1.02	0.92	0.037	0.173
SOD (U/mg protein)	158.02	161.10	155.40	2.486	0.696

^a, b^ Means within a row with different superscripts differ significantly at *p* < 0.05. ^1^ MDA, malondialdehyde; T-SOD, total superoxide dismutase; T-AOC, total antioxidant capacity. ^2^ CON: broilers receiving a corn-soybean meal basal diet without antibiotic and zeolite; ANT: broilers receiving the basal diet supplemented with 50 mg/kg chlortetracycline; ZEO: broilers were given the basal diet supplemented with 10 g/kg zeolite. SEM, standard error of means (n = 6).

**Table 4 animals-09-00909-t004:** Effects of zeolite and antibiotic supplementation on intestinal barrier function in broilers.

Items ^1,2^	CON	ANT	ZEO	SEM	*p*-Value
21 days					
DAO (U/mL)	18.80	17.87	18.37	0.980	0.932
LPS (EU/L)	0.53 ^a^	0.45 ^ab^	0.32 ^b^	0.032	0.014
D-lactic acid (mmol/L)	2.22	2.20	2.64	0.154	0.450
42 days					
DAO (U/mL)	25.38 ^a^	25.03 ^a^	17.11 ^b^	1.150	<0.001
LPS (EU/L)	0.32 ^ab^	0.38 ^a^	0.26 ^b^	0.019	0.017
D-lactic acid (mmol/L)	3.83 ^a^	3.64 ^a^	2.64 ^b^	0.186	0.009

^a, b^ Means within a row with different superscripts differ significantly at *p* < 0.05. ^1^ DAO, diamine oxidase; LPS, bacterial lipopolysaccharide. ^2^ CON: broilers receiving a corn-soybean meal basal diet without antibiotic and zeolite; ANT: broilers receiving the basal diet supplemented with 50 mg/kg chlortetracycline; ZEO: broilers were given the basal diet supplemented with 10 g/kg zeolite. SEM, standard error of means (n = 6).

**Table 5 animals-09-00909-t005:** Effect of zeolite and antibiotic supplementation on intestinal mucosal gene expression levels related to barrier function in broilers.

Items ^1,2^	CON	ANT	ZEO	SEM	*p*-Value
21 days					
Duodenum					
*OCLN*	1.00	0.83	0.79	0.061	0.327
*CLDN2*	1.00 ^a^	0.61 ^b^	1.01 ^a^	0.077	0.046
*CLDN3*	1.00	0.97	1.38	0.140	0.459
*ZO-1*	1.00	0.85	0.91	0.061	0.646
Jejunum					
*OCLN*	1.00	1.02	1.02	0.056	0.985
*CLDN2*	1.00	0.91	0.94	0.090	0.932
*CLDN3*	1.00	0.94	0.99	0.073	0.937
*ZO-1*	1.00 ^b^	0.95 ^b^	1.52 ^a^	0.080	0.001
Ileum					
*OCLN*	1.00	1.07A	1.14	0.133	0.923
*CLDN2*	1.00	0.96	0.90	0.105	0.936
*CLDN3*	1.00	0.91	1.20	0.199	0.843
*ZO-1*	1.00	0.87	0.99	0.059	0.653
42 days					
Duodenum					
*OCLN*	1.00 ^b^	1.33 ^ab^	1.71 ^a^	0.113	0.030
*CLDN2*	1.00	1.02	1.09	0.071	0.864
*CLDN3*	1.00	1.05	1.38	0.189	0.703
*ZO-1*	1.00	0.97	1.04	0.064	0.931
Jejunum					
*OCLN*	1.00	0.92	1.06	0.077	0.788
*CLDN2*	1.00	1.06	1.11	0.133	0.947
*CLDN3*	1.00	1.19	1.27	0.161	0.801
*ZO-1*	1.00	1.09	1.16	0.097	0.821
Ileum					
*OCLN*	1.00	1.04	0.81	0.046	0.090
*CLDN2*	1.00	1.05	1.29	0.228	0.875
*CLDN3*	1.00	0.69	0.74	0.059	0.057
*ZO-1*	1.00	0.96	1.06	0.065	0.861

^a, b^ Means within a row with different superscripts differ significantly at p < 0.05. ^1^
*CLDN2*, claudin-2; *CLDN3*, claudin-3; *OCLN*, occludin; *ZO-1*, zonula occludens-1. ^2^ CON: broilers receiving a corn-soybean meal basal diet without antibiotic and zeolite; ANT: broilers receiving the basal diet supplemented with 50 mg/kg chlortetracycline; ZEO: broilers were given the basal diet supplemented with 10 g/kg zeolite. SEM, standard error of means (n = 6).

**Table 6 animals-09-00909-t006:** Effects of zeolite and antibiotic supplementation on detection frequency (DF) of the 11 targeted *tet* genes in cecum samples of broilers (%).

Items ^1^	21 Days				42 Days			
	CON	ANT	ZEO	DF	CON	ANT	ZEO	DF
*TetA*	0	0	16.67	8.33	0	0	16.67	4.16
*TetB*	50.00	100.00	100.00	87.50	100.00	100.00	100.00	100.00
*TetC*	33.33	100.00	100.00	83.33	100.00	100.00	100.00	100.00
*TetE*	16.67	83.33	50.00	50.00	33.33	50.00	16.67	37.50
*TetG*	100.00	100.00	100.00	100.00	50.00	100.00	66.67	66.67
*TetM*	100.00	100.00	100.00	100.00	100.00	100.00	100.00	100.00
*TetO*	100.00	100.00	100.00	100.00	100.00	100.00	100.00	100.00
*TetQ*	100.00	100.00	100.00	100.00	100.00	100.00	100.00	100.00
*TetT*	50.00	66.67	16.67	45.83	50.00	50.00	83.33	58.33
*TetW*	100.00	100.00	100.00	100.00	100.00	100.00	100.00	100.00
*TetX*	83.33	100.00	100.00	95.83	100.00	100.00	100.00	100.00
DF	66.67	86.36	80.30	79.17	75.76	81.82	80.30	78.79

^1^ CON: broilers receiving a corn-soybean meal basal diet without antibiotic and zeolite; ANT: broilers receiving the basal diet supplemented with 50 mg/kg chlortetracycline; ZEO: broilers were given the basal diet supplemented with 10 g/kg zeolite.

**Table 7 animals-09-00909-t007:** Effects of zeolite and antibiotic supplementation on relative abundances of ARGs in cecum samples of broilers.

Items ^1^	CON	ANT	ZEO	SEM	*p*-Value
21 days					
*TetB*	1.88 × 10^−5^	5.09 × 10^−5^	7.97 × 10^−5^	3.65 × 10^−5^	0.321
*TetC*	1.42 × 10^−5^	2.95 × 10^−5^	3.20 × 10^−5^	5.66 × 10^−6^	0.666
*TetG*	4.34 × 10^−5^	6.98 × 10^−5^	5.23 × 10^−5^	7.23 × 10^−6^	0.335
*TetM*	9.34 × 10^−4^	2.01 × 10^−3^	6.46 × 10^−4^	2.50 × 10^−4^	0.055
*TetO*	1.96 × 10^−1^	1.82 × 10^−1^	2.02 × 10^−1^	2.28 × 10^−2^	0.942
*TetQ*	1.97 × 10^−1^	2.18 × 10^−1^	1.48 × 10^−1^	3.47 × 10^−2^	0.722
*TetW*	4.86 × 10^−1^	6.02 × 10^−1^	5.12 × 10^−1^	3.12 × 10^−2^	0.294
*TetX*	8.67 × 10^−6^	2.41 × 10^−5^	3.45 × 10^−5^	6.43 × 10^−6^	0.296
42 days					
*TetB*	3.64 × 10^−5 b^	1.95 × 10^−4 a^	1.68 × 10^−5 b^	2.41 × 10^−5^	<0.001
*TetC*	2.49 × 10^−5 ab^	1.20 × 10^−4 a^	9.80 × 10^−6 b^	1.99 × 10^−5^	0.034
*TetG*	1.67 × 10^−5^	3.40 × 10^−5^	1.72 × 10^−4^	3.67 × 10^−5^	0.202
*TetM*	1.76 × 10^−3^	2.33 × 10^−3^	2.59 × 10^−3^	3.42 × 10^−4^	0.628
*TetO*	2.64 × 10^−1^	2.99 × 10^−1^	2.10 × 10^−1^	2.59 × 10^−2^	0.393
*TetQ*	3.54 × 10^−1^	8.07 × 10^−1^	6.40 × 10^−1^	9.04 × 10^−2^	0.114
*TetW*	6.47 × 10^−1^	9.06 × 10^−1^	6.43 × 10^−1^	5.23 × 10^−2^	0.052
*TetX*	2.90 × 10^−1 ab^	5.74 × 10^−1 a^	1.57 × 10^−1 b^	7.13 × 10^−2^	0.039

^1^ CON: broilers receiving a corn-soybean meal basal diet without antibiotic and zeolite; ANT: broilers receiving the basal diet supplemented with 50 mg/kg chlortetracycline; ZEO: broilers were given the basal diet supplemented with 10 g/kg zeolite. SEM, standard error of means (n = 6).

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
