# Peer review of "Effects of Dietary Zeolite Supplementation as an Antibiotic Alternative on Growth Performance, Intestinal Integrity, and Cecal Antibiotic Resistance Genes Abundances of Broilers"

_animals, 2019, doi:10.3390/ani9110909_

Round 1
Reviewer 1 Report
tHis word http://www.jstage.jst.go.jp/browse/jpsa doi:10. 2141/ jpsa.016007Author Response
Response to Reviewer 1 Comments
Thank you so much for your reviews on our manuscript. However, we didn't find the comments and suggestions. And we have revised this original paper carefully according to the comments by the other two reviewers. We sincerely hope that the corrections in the revised manuscript would be sufficient to make our manuscript suitable for publication.

Reviewer 2 Report
the manuscript contains some interesting results, but several improvements need before consider it for publication.
Line 52. Please, consider also the manuscript "Mercurio M., Cappelletti P., De Gennaro B, De Gennaro M., Bovera F., Iannaccone F., Grifa C., Langella A, Moretti V., Esposito L. 2016. The effect of digestive activity of pig gastro-intestinal tract on zeolite-rich rocks: an in vitro study. Microsporous and Mesosporous Materials 225, pp. 133-136.
line 68: put a comma between 144 and 1
lines 71-76: please, rewrite. Avoid to use too much ":" and use similar times of the verbs
line 75: indicate why the use of 10 g/kg of zeolite and why it is not necessary to test more doses?
line 70: why the birds were raised in cages? The size of the cages must be indicated. What is the sex ratio?
line 93: min-duodenum?
line 161: delete "significantly". Delete "whereas...21 days"
line 167: delete "with..(P<0.05)"
lines 170-171: delete "were...p<0.05". "Numerically improved" has not sense considering the statistical point of view
lines 203-205: delete as it is an introduction and not a discussion
Author Response
Response to Reviewer 2 Comments
Point 1: Line 52. Please, consider also the manuscript "Mercurio M., Cappelletti P., De Gennaro B, De Gennaro M., Bovera F., Iannaccone F., Grifa C., Langella A, Moretti V., Esposito L. 2016. The effect of digestive activity of pig gastro-intestinal tract on zeolite-rich rocks: an in vitro study. Microsporous and Mesosporous Materials 225, pp. 133-136.
Response 1: Thank you for this kind advice. This recommended reference has been added as required, sir. (Reference [20]. Line 52, Line 385-386)
Point 2: line 68: put a comma between 144 and 1
Response 2: Thank you very much! We have added comma between “144” and “1” according to your comment, sir. (Line 70)
Point 3: lines 71-76: please, rewrite. Avoid to use too much ":" and use similar times of the verbs
Response 3: Thanks for this constructive comment! This section has been totally rewritten according to your beneficial advice, sir. (Line 72-76).
Point 4: line 75: indicate why the use of 10 g/kg of zeolite and why it is not necessary to test more doses?
Response 4: Your insightful comments are highly appreciated, sir! The maximum dose of zeolite additive in animal feed in the official legislation document issued by European Union is 20 g/kg [1]. An in vivo study on broilers has showed that dietary zeolite supplementation at a level of 10 g/kg has similar effects to its dosage at 20 g/kg [2]. Moreover, studies have also showed that 10 g/kg zeolite supplementation exhibits beneficial consequences in improving growth performance and intestinal flora composition of broilers or laying hens. The similar results have been also reported previously [3, 4]. According to these aforementioned findings, sir. We therefore supplemented 10 g/kg zeolite in this study, and we have clarified this point in the revised manuscript as required. (Line 76-78)
[1]EU commission. Commission regulation (EU) 575/2011 of 16 June 2011 on the catalogue of feed materials. Off. J. Eur. Union, 2011, 159, 25-65.
[2]Mallek, Z.; Fendri, I.; Khannous, L.; Hassena, A. B.; Traore, A. I.; Ayadi, M. A.; Gdoura, R. Effect of zeolite (clinoptilolite) as feed additive in Tunisian broilers on the total flora, meat texture and the production of omega 3 polyunsaturated fatty acid. Lipids Health Dis. 2012, 11, 35.
[3]Suchý, P.; Strakova, E.; Večerek, V.; Klouda, Z.; Kráčmarová, E. The effect of a clinoptilolite-based feed supplement on the performance of broiler chickens. Czech J. Anim. Sci. 2006, 51, 168-173.
[4]Elliot, M. A.; Edwards JR, H. M. Comparison of the effects of synthetic and natural zeolite on laying hen and broiler chicken performance. Poult. Sci. 1991, 70, 2115-2130.
Point 5: line 70: why the birds were raised in cages? The size of the cages must be indicated. What is the sex ratio?
Response 5: Thank you, sir! The cage breeding is a common practice of broilers production in Asian especially in China, Africa and parts of eastern Europe [1]. It can promote growth, save space, prevent disease infection, and facilitate management. The cage feeding is also available in abundant literature [2-5]. The cage space and sex ratio have been provided as required, sir. (Line 80)
[1]Shields, S.; Greger, M. Animal welfare and food safety aspects of confining broiler chickens to cages. Animals. 2013, 3, 386-400.
[2]Wu, Q. J.; Wang, L. C.; Zhou, Y. M.; Zhang, J. F.; Wang, T. Effects of clinoptilolite and modified clinoptilolite on the growth performance, intestinal microflora, and gut parameters of broilers. Poult. Sci. 2013, 92, 684-692.
[3]Cheng, Y. F.; Chen, Y. P.; Chen, R.; Su, Y.; Zhang, R. Q.; He, Q. F.; Zhou, Y. M. Dietary mannan oligosaccharide ameliorates cyclic heat stress-induced damages on intestinal oxidative status and barrier integrity of broilers. Poult. Sci. 2019. DOI: 10.3382/ps/pez192
[4]Okeke, O. R.; Ujah, I. I.; Okoye, P. A. C.; Ajiwe, V. I. E.; Eze, C. P. Effect of different levels of cadmium, lead and arsenic on the growth performance of broiler and layer chickens. J. Applied Chem. 2015, 8, 57-59.
[5]Zhou, P.; Tan, Y. Q.; Zhang, L.; Zhou, Y. M.; Gao, F.; Zhou, G. H. Effects of dietary supplementation with the combination of zeolite and attapulgite on growth performance, nutrient digestibility, secretion of digestive enzymes and intestinal health in broiler chickens. Asian-Australas. J. Anim. Sci. 2014, 27, 1311.
[6]Rikimaru, K.; Takeda, H.; Uemoto, Y.; Komatsu, M.; Takahashi, D.; Suzuki, K.; Takahashi, H. Effect of a single-nucleotide polymorphism in the cholecystokinin type A receptor gene on growth traits in the Hinai-dori chicken breed. J. Poult. Sci. 2013, 0120130.
Point 6: line 93: min-duodenum?
Response 6: Thank you for the patience. We are sorry for the wrong use of "mid-duodenum", which has been rewritten in the revised manuscript. (Line 97)
Point 7: line 161: delete "significantly". Delete "whereas...21 days"
Response 7: Thanks! We have deleted them as required. (Line 167)
Point 8: line 167: delete "with..(P<0.05)"
Response 8: Thanks! We have deleted it in the revised manuscript. (Line 174)
Point 9: lines 170-171: delete "were...p<0.05". "Numerically improved" has not sense considering the statistical point of view
Response 9: Thank you for this insightful comment. These sentences have been deleted as required. (Line 177)
Point 10: lines 203-205: delete as it is an introduction and not a discussion
Response 10: Thank you for insightful comments! We have corrected this part as required, sir. (Line 209)

Reviewer 3 Report
Zeolite is a microporous, aluminosilicate mineral. What’s its mechanism to improve growth and gut development in birds? In this trial, both the growth performance and morphology results showed that the zeolite supplementation had significant improvement than the antibiotic treatment. Any explanation and mechanism for these results? Line 15, what do you mean “cacel”? Correct it in the title as well. Line 22, add doses for antibiotic and zeolite in the abstract. Line 76-77: what’s the feed form? Line 86, the experimental design included 3 treatments and 6 replicates per treatment. How did the author select the 24 birds from the total 18 cages? As the author mentioned, if one bird per cage was selected, the total sampling number should be 18? Line 100-105: how many slides reading for each intestinal segment sample? Line 197-201: Did the author have any reason or hypothesis to find the changes of the ARGs for the ZEO treatment? Why did author think measuring the ARGs for the ZEO treatment is important in current experimental design? Line 202, the author listed several references related to the zeolite studies in broiler chickens. However, further declarations and reference articles need to be added in the discussion section to explain the potential mechanism and theory behind such improvement on both growth and intestinal development. Line 205-209: It’s bold to draw this conclusion from a single animal trial and based on the limited number of animals. Line 275: For the zeolite, it has the potential to absorb the harmful substances. How does the zeolite kill the harmful bacteria? Again, it’s not appropriate to draw such conclusion based on a single animal experiment.Author Response
Point 1: Zeolite is a microporous, aluminosilicate mineral. What’s its mechanism to improve growth and gut development in birds? In this trial, both the growth performance and morphology results showed that the zeolite supplementation had significant improvement than the antibiotic treatment. Any explanation and mechanism for these results?
Response 1: Thank you for this comment. And we have provided corresponding revisions in the revised manuscript (Line 206-212). The mechanism of zeolite to improve growth and gut development is presented as follows, Firstly, ammonia binding effect of zeolite can relieve the toxic effects of ammonia produced by intestinal microflora [1, 2]. Secondly, zeolite can reduce the absorption of toxic products generated during intestinal microbial degradation, such as p-cresol [3]. Thirdly, zeolite can prolong the retention time of intestinal digesta and therefore improve the utilization of nutrients [4, 5]. Fourthly, pancreatic enzymes activity can be improved by fed with zeolite [6]. Lastly, zeolite can alleviate growth inhibitory effects of mycotoxins [7]. These explanations have been provided in the corresponding content in the revised manuscript as required, sir. (Line 211-217)
[1]Shurson, G. C.; Ku, P. K.; Miller, E. R.; Yokoyama, M. T. Effects of zeolite or clinoptilolite in diets of growing swine. J. Anim. Sci. 1984, 59, 1536-1545.
[2]Pond, W. G.; Yen, J. T.; Varel, V. H. Response of growing swine to dietary copper and clinoptilolite supplementation. Nutr. Rep. Int. (USA). 1988, 37, 795.
[3]Pond, W. G.; Mumpton, F. A. Zeo-agriculture: Use of natural zeolites in agriculture and aquaculture. Westview press. 1984.
[4]Mumpton, F. A.; Fishman, P. H. The application of natural zeolites in animal science and aquaculture. J. Anim. Sci. 1977. 45, 1188-1203.
[5]Olver, M. D. Effect of feeding clinoptilolite (zeolite) on the performance of three strains of laying hens. Brit. Poultry Sci. 1997, 38, 220-222.
[6]Zhou, P.; Tan, Y. Q.; Zhang, L.; Zhou, Y. M.; Gao, F.; Zhou, G. H. Effects of dietary supplementation with the combination of zeolite and attapulgite on growth performance, nutrient digestibility, secretion of digestive enzymes and intestinal health in broiler chickens. Asian-Australas. J. Anim. Sci. 2014, 27, 1311.
[7]Harvey, R. B.; Kubena, L. F.; Elissalde, M. H.; Phillips, T. D. Efficacy of zeolitic ore compounds on the toxicity of aflatoxin to growing broiler chickens. Avian Dis. 1993, 37, 67-73.
Point 2: Line 15, what do you mean “cacel”? Correct it in the title as well. Line 22, add doses for antibiotic and zeolite in the abstract.
Response 2: Thank you very much! We have corrected them according to the comments. (Line 4, 15)
Point 3: Line 76-77: what’s the feed form?
Response 3: The feed form is mash. We have actually declared the mash feed form in the original manuscript, sir. (Line 80)
Point 4: Line 86, the experimental design included 3 treatments and 6 replicates per treatment. How did the author select the 24 birds from the total 18 cages? As the author mentioned, if one bird per cage was selected, the total sampling number should be 18?
Response 4: Yes, the total number of broilers for sampling should be 18. We are so sorry this, and have corrected this point, sir. (Line 89)
Point 5: Line 100-105: how many slides reading for each intestinal segment sample?
Response 5: Thank you! Histological slides were prepared from 3 cross-sections (5 µm thick) of each intestinal sample. A total of 10 intact, well-oriented crypt-villus and associated crypts were measured from each segment. (Line 107-108)
Point 6: Line 197-201: Did the author have any reason or hypothesis to find the changes of the ARGs for the ZEO treatment?
Response 6: Sir, studies have showed that natural zeolite could reduce the environmental risks of ARGs in sludge compost. This may due to its sporous structure and the ability to reduce the selective pressure from heavy metals and the rate of microbe contact, and then horizontal gene transfer through conjugation is reduced [1]. However, whether zeolite could reduce ARGs in animal body remain unknown. Therefore, we hypothesized zeolite inclusion may reduce ARGs in vivo when supplementing it to broiler feed. And we have added the necessary elucidation as required, sir. (Line 282-284)
[1]Zhang, J.; Chen, M.; Sui, Q.; Tong, J.; Jiang, C.; Lu, X.; Zhang, Y.; Wei, Y. Impacts of addition of natural zeolite or a nitrification inhibitor on antibiotic resistance genes during sludge composting. Water Res. 2016, 91, 339-349.
[2]Zhang, J.; Sui, Q.; Zhong, H.; Meng, X.; Wang, Z.; Wang, Y.; Wei, Y. Impacts of zero valent iron, natural zeolite and Dnase on the fate of antibiotic resistance genes during thermophilic and mesophilic anaerobic digestion of swine manure. Bioresour. Technol. 2018, 258, 135-141.
[3]Peng, S.; Li, H.; Song, D.; Lin, X.; Wang, Y. Influence of zeolite and superphosphate as additives on antibiotic resistance genes and bacterial communities during factory-scale chicken manure composting. Bioresour. Technol. 2018, 263, 393-401.
Point 7: Why did author think measuring the ARGs for the ZEO treatment is important in current experimental design?
Response 7: ARGs, as a new type of environmental pollutant and a threat of damaging public’s health, has attracted wide attention [1, 2]. Antibiotic usage in poultry production has lasted decades, which has resulted in antibiotic residue and ARGs occurrence [3]. In vitro studies showed that zeolite can adsorb antibiotic and ARGs [4, 5]. Poultry are currently perceived as one of major sources of pathogens and antimicrobial resistance [6]. However, whether zeolite can reduce ARG produced during the process of growth and development in broilers are still unknown. We therefore detect ARGs abundance in cecal content to investigate whether zeolite has such effect in vivo, and if so, which can provide possibility to reduce threat both public health and environment pollution. (Line 57-60)
[1]Pruden, A.; Pei, R.; Storteboom, H.; Carlson, K. H. Antibiotic resistance genes as emerging contaminants: Studies in northern Colorado. Environ. Sci. Technol. 2006, 40, 7445-7450.
[2]Rysz, M.; Alvarez, P. J. Amplification and attenuation of tetracycline resistance in soil bacteria: Aquifer column experiments. Water Res. 2004, 38, 3705-3712.
[3]Singer, R. S.; Finch, R.; Wegener, H. C.; Bywater, R.; Walters, J.; Lipsitch, M. Antibiotic resistance-the interplay between antibiotic use in animals and human beings. Lancet Infect. Dis. 2003, 3, 47-51.
[4]de Sousa, D. N. R.; Insa, S.; Mozeto, A. A.; Petrovic, M.; Chaves, T. F.; Fadini, P. S. Equilibrium and kinetic studies of the adsorption of antibiotics from aqueous solutions onto powdered zeolites. Chemosphere. 2018, 205, 137-146.
[5]Zhang, J.; Chen, M.; Sui, Q.; Tong, J.; Jiang, C.; Lu, X.; Zhang, Y.; Wei, Y. Impacts of addition of natural zeolite or a nitrification inhibitor on antibiotic resistance genes during sludge composting. Water Res. 2016, 91, 339-349.
[6]Prasai, T. P.; Walsh, K. B.; Bhattarai, S. P.; Midmore, D. J.; Van, T. T.; Moore, R. J.; Stanley, D. Zeolite food supplementation reduces abundance of enterobacteria. Microbiol. Res. 2017, 195, 24-30.
Point 8: Line 202, the author listed several references related to the zeolite studies in broiler chickens. However, further declarations and reference articles need to be added in the discussion section to explain the potential mechanism and theory behind such improvement on both growth and intestinal development.
Response 8: Thank you for these insightful comments. We have provided necessary articles and revised these sentences in our manuscript (Line 211-217).
Point 9: Line 205-209: It’s bold to draw this conclusion from a single animal trial and based on the limited number of animals.
Response 9: Thank you! This point has been revised together with the last question (point 8).
Point 10: Line 275: For the zeolite, it has the potential to absorb the harmful substances. How does the zeolite kill the harmful bacteria?
Response 10: Zeolite may have a direct inhibition of bacteria, but it is unclear whether zeolite could kill pathogenic bacteria [1]. There are several possible reasons:
Zeolite could adsorb bacterial species, and then alter metabolism and inhibit the growth of bacteria, ultimately leading to bacteria death [2]. Zeolite can reduce moisture of the environment that bacteria live in, causing a reduction of water activity that decreases bacterial viability [3].[1]Ricke, S. C.; Pillai, S. D.; Widmer, K. W.; Ha, S. D. Survival of Salmonella typhimurium in soil and liquid microcosms amended with clinoptilolite compounds. Bioresour. technol. 1995, 53, 1-6.
[2]Shurson, G. C.; Ku, P. K.; Miller, E. R.; Yokoyama, M. T. Effects of zeolite or clinoptilolite in diets of growing swine. J. Anim. Sci. 1984, 59, 1536-1545.
[3]Flamingen, E. M.; Mumpton, F. A. Commercial properties of natural zeolites. Rev. Mineral. 1981, 4, 165-75.
Point 11: it’s not appropriate to draw such conclusion based on a single animal experiment.
Response 11: Thanks for this constructive comment! The “Conclusion” section has been revised as required, and we sincerely wish this revision would be ok, sir. (Line 291-293)

Round 2
Reviewer 2 Report
dear authors, thank you for changing your manuscript
Reviewer 3 Report
Accept for publication